Evidence synthesis

environmental science/pedology

soil carbon, carbon sequestration, soil ecosystem services, re-carbonization, soil carbon economics, payment

**Author for correspondence:**
Brian J. Reid
e-mail: b.reid@uea.ac.uk

A contribution to the 'Sustainable Land Use' special collection.

# Capturing a soil carbon economy

Sam G. Keenor[1], Aline F. Rodrigues[2,3], Li Mao[1], Agnieszka E. Latawiec[1,2,3,4], Amii R. Harwood[1] and Brian J. Reid[1]

[1]School of Environmental Sciences, University of East Anglia, Norwich Research Park, Norwich, UK
[2]Department of Geography and the Environment, Rio Conservation and Sustainability Science Centre, Pontifical Catholic University of Rio de Janeiro, Rio de Janeiro, Brazil
[3]International Institute for Sustainability, Rio de Janeiro, Brazil
[4]Department of Production Engineering, Logistics and Applied Computer Science, Faculty of Production and Power Engineering, University of Agriculture, Kraków, Poland

SGK, 0000-0001-8620-8133

Current carbon pricing and trading mechanisms, despite their efficacy in reducing GHG emissions from industry, will not be sufficient to achieve Net Zero targets. Current mechanisms that redress emissions are largely economic *disincentives*, in effect financial penalties for emitters. In order to attain Net Zero futures, financial *incentives* for activities that sequester carbon from the atmosphere are needed. Herein, we present the environmental and economic co-benefits of soil re-carbonization and justify support for soil carbon remuneration. With increasing momentum to develop green economies, and projected increases in carbon price, growth in the global carbon market is inevitable. The establishment of a soil-based carbon economy, within this emerging financial space, has the potential to deliver a paradigm shift that will accelerate climate change mitigation, and concurrently realize net gains for soil health and the delivery of soil ecosystem services. Pivotal to the emergence of a global soil carbon economy will be a consensus on certification instruments used for long-term soil carbon storage, and the development of robust institutional agreements and processes to facilitate soil carbon trading.

## 1. Introduction

Soils support all life on Earth. They provide a primary source of food and resources, filter water, regulate climate and provide the strata on which terrestrial life is supported [1]. The prosperity and economic status of nations are inextricably linked to the health of soils [2,3]. Yet to many, soil is dirt, a nuisance and unclean. This mentality of 'inconvenience' has contributed to the

damage and degradation of one of the most precious, largely non-renewable resources on Earth [4,5]. Furthermore, this myopia precludes appreciation that soils are, in fact, living, dynamic and essential ecosystems, providing not only tangible 'goods', but also services that support, regulate and sustain the global system [3,6,7].

Overt linkages connect the climate system to soil-mediated regulation of climate-relevant atmospheric gases. In particular, soils play a fundamental role in the two-way exchange of carbon (as $CO_2$ and $CH_4$) and nitrogen (as $N_2$, $N_2O$ and $NH_3$) [8,9]. Soil carbon is central to shaping edaphic soil factors (§2) [10]. This carbon facilitates soil aggregation, development of soil structure; and thus, the physical flows of water and gases [8]. The loss of soil carbon, via mineralization to $CO_2$ and/or erosion, results in a reduction of the soil carbon stock, thereby increasing atmospheric concentrations of carbon (primarily $CO_2$), and/or undermining the integrity of the soil across its inextricably linked chemical, biological and physical attributes [11]. Soil degradation has wide-reaching consequences for biodiversity, food security, freshwater provision and wider ecosystem service delivery [12–15]. It is emphasized that damage done to soil is not confined to soil; it has negative impacts on the entire planetary system (§3).

Strategies and tools are urgently needed to combat both soil degradation and climate change [16]. Facilitating a method of economic remuneration for re-carbonization of soils has a potential to act beneficially on both counts (§4). In this paper, we explain the pivotal importance of soil carbon and the fundamental role it plays in sustaining the delivery of key ecosystem services. We explain the premise and operation of carbon markets (§5), evaluate how these may be aligned to realize policy (§6) and propose a platform/mechanism that will allow payments to be collected and divested to re-carbonize soils (§7). Thereafter, we discuss issues pertaining to carbon permanence, and the barriers that must be overcome to deliver a trading platform that supports a soil carbon economy (§8).

# 2. The indispensability of soil carbon

Due to differences in climate, parent material and formation conditions, soils vary greatly across the surface of the Earth. Soils are dynamic and complex matrices; composed of organic and inorganic materials, water, air and organisms [3], with each constituent contributing to the effective functioning of the wider soil system. Soils provide many valuable ecosystem services [14,17,18]. Specifically, soils facilitate 'goods' and service provisions, such as resource and food productions (*provisioning services*), water filtration, flood mitigation and climate regulation (*regulating services*), carbon sequestration and carbon storage (*supporting services*), and aesthetics and recreation (*cultural services*) [3,6,16,19]. The health of a soil is categorized by its capacity to sustain life, and the extent to which it may enhance or maintain the provision of ecosystem services [20,21].

Healthy soils show greater resistance to stress [22,23], providing greater resilience to the negative impacts of drought, flood and erosion [24]. Of primary importance to soil health is soil carbon [23,25], and the ecosystem services it sustains [19,20,26–28]. Soil carbon exerts influence over a variety of soil attributes, including physical, chemical, hydrological and biological properties [22,29–32]. Thus, soil carbon is a robust proxy with which to gauge soil health and quality.

Soil organic matter (SOM), comprising organic forms of carbon and other bioactive elements (nitrogen, phosphorus, sulfur), is derived from the remnants of plant, animal and microbial material. SOM contains both labile and recalcitrant fractions, in different stages of decomposition and decay [23,25,33,34]. Soils naturally sequester carbon through the accumulation of dead and decaying organic matter that is slowly incorporated and stored [9,23,25]. These different forms of SOM provide the resource to prime soil life (via the labile carbon pool that can be used relatively easily), and the means to deliver long-term carbon storage (via the recalcitrance carbon pool that resists degradation) [35,36].

Soils with the high organic matter have a more developed soil structure, with greater aggregation and cohesion [29,37]. These structures are more resistant to drought and erosion, due to improved porosity and reduced compaction [24]. Well-aggregated and well-structured soils are more accommodating to rainfall [38]. Thus, improving water infiltration, water storage and buffering of the hydrological cycle [39]. In addition, more developed soil aggregates provide stronger physical protection to SOM stocks [40].

Globally, soils contain 2000–2500 Pg C; thus, soils hold approximately three times more carbon than the atmosphere [13,41,42]. This soil carbon store is not fixed or permanent; in reality, it is in dynamic equilibrium with other Earth systems [13,43]. Changes in land use (e.g. forest versus pasture versus arable) greatly alter the balance of carbon stored in soil and in the atmosphere [20,44]. Consequently, actions that alter land use also alter soil carbon stocks, influence atmospheric carbon levels and, thus by extension, the global climate system [43].

# 3. Decarbonization of soil

Damage caused to the soil system through anthropogenic action has occurred at an unprecedented rate. In the last 150 years, more than half of all soils have been damaged [45]. Degradation of soil has been accompanied by the attrition of greater than 50% of the soil organic carbon (SOC) stock in some cultivated soils, with over 2 billion ha affected globally [12,13,46]. Soils subjected to degradation become a significant emission source of $CO_2$ to the atmosphere [4,12,47]. Soil degradation has liberated an estimated 176 Gt of soil carbon globally [48]; a significant quotient when contextualized against the 800 Gt C held in the atmosphere [41]. Averaged over the last 150 years, the soil carbon loss rate equates to $1.6 \pm 0.8$ Gt C $yr^{-1}$ [44]. In context, anthropogenic global carbon emissions in 2000 were estimated to be 7.5 Gt C $yr^{-1}$ [49] (i.e. the rate of annual SOC loss is approx. 20% of this value). Agriculture, forestry and land use change is reported to be directly responsible for approximately 18–24% of total anthropogenic GHG emission each year [43,50]. This conversion of natural ecosystems to managed systems is reported to deplete SOC stocks by an average of 60% in temperate regions, and up to 75% in the worst affected regions of the tropics, accounting for losses of up to 80 t C $ha^{-1}$ [13].

Inadequate SOC stocks have been linked to impaired soil function, reduced nutrient provision and water availability, and loss of below- and above-ground biodiversity [46,51,52]. SOM degradation increases the vulnerability of soils to erosion and accelerates the desertification process [53,54]. It is important to appreciate that soil resources, although abundant and long lasting, are non-renewable on an anthropogenic timescale [5]. Where rates of soil loss/degradation outpace rates of biogenic and geological soil replacement/recovery, the sustainability balance is tipped [26,48,55]. Globally, poor soil management and loss of SOC have exacerbated topsoil losses to a point where they are 10–40 times greater than natural replacement rates: in the USA, topsoil loss rates are roughly 10 times that of replacement; while in India and China, loss rates exceed 30–40 times natural replacement [56].

Failures in soil management decrease crop yields [57,58] and impair society's ability to grow sufficient crops [5,59]. Degraded SOC stocks have been reported to underpin decreases in crop productivity of 0.3% per year; a decrease, which if not arrested, may aggregate to an average of 10% reduction in yields by 2050 (with the worst affected regions experiencing up to 50% yield reductions) [4,48]. Across the European Union, 45% of agricultural soils are considered impaired or very impaired in SOM content [60].

# 4. Re-carbonization of soil

The agricultural sector has potential to transition from a significant net source of GHG emissions to a net carbon sink [61,62]. By altering land/soil management practices, the negative effects of agriculture upon soils and the environment may be substantially abated [25,31,63,64]. Agricultural soils have potential to make significant contributions to carbon capture and storage in both the long and short term [12,13,31,65,66].

Taking the UK as an example, emissions of GHG from agricultural sources in 2017 were 45.6 million tonnes $CO_2e$ ($CO_2e$ = total global warming potential of all emissions normalized to $CO_2$ temperature forcing potential [64]), delivering 1/10 of the total UK emission (435.2 Mt $CO_2e$ (2019) [67]). It is highlighted that agricultural GHG emissions differ from those associated with industries such as fossil fuel energy. In contrast with these industries (that emit predominantly $CO_2$), agricultural sector emissions are, for the most part, associated with $CH_4$ and $N_2O$, accounting for up to 80% of total agricultural emission [68,69]. In the UK, total agricultural emissions are split: 40% $CH_4$ and 50% $N_2O$ and 10% $CO_2$ [70,71]. The UK National Farmers' Union (NFU), the largest farmers' organization, has suggested three pillars of intervention to offset the majority of agricultural GHG [70]. These pillars relate to: (1) improving farming productive efficiency; (2) farmland carbon storage; and (3) boosting renewable energy and the wider bio-economy. Under pillar 2, the NFU Aspiration seeks to sequester 9 Mt $CO_2e$ $yr^{-1}$. Most of this carbon capture is linked with interventions that enhance soil carbon storage (5 Mt $CO_2e$ $yr^{-1}$), and peatland and wetland restoration (3 Mt $CO_2e$ $yr^{-1}$). Taken together, these interventions are projected to deliver approximately 20% offset against agricultural sector GHG emission in the UK by 2040.

*Carbon farming* (box 1) describes the holistic approach of using agricultural methods to reduce or offset GHG emission from agriculture; through the capture and storage of carbon in soils and vegetation [92]. Increasing the carbon stock of soils, on a global scale, has an estimated sequestration potential of 3.4–5 Gt C $yr^{-1}$ [42,66,93].

Effective methods for significantly increasing SOC stocks, within a short time frame, include lower impact tillage approaches and the use of soil amendments, such as compost, paper crumble, manure

**Box 1.** Carbon farming—case study Australia.

To reduce emissions and meet government commitments (80% emission reductions from 2000 levels by 2050), Australia adopted a national carbon pricing mechanism (CPM) in 2011. This was facilitated through the creation of an Australian emissions trading scheme (ETS) (that covered approx. 50% of national emissions from a range of sectors (excluding agriculture)), and increases in fuel duties [72–74]. To run concurrently with this ETS, the carbon farming initiative (CFI) was adopted to provide offsets that could be used within, and promote emissions reduction within, the agricultural sector [72,73,75]. The CFI was supported by the Australian Carbon Pricing Scheme and issued carbon credit units for each tonne of $CO_2$e abated or sequestered [73,76–79]. The CFI was the first nationwide example of carbon credit creation and trade by the agriculture and forestry sectors to a wider market [77,80]. Carbon farming methods pertained to activities that increase soil carbon stocks and/or store carbon within vegetation, or facilitated emissions avoidance [73,81]. Accepted methods of carbon sequestration under the CFI included: limiting inputs of agrochemicals (e.g. inorganic fertilizers) to the soil, limiting the use of aggressive tillage regimes (transition to minimal/no till), implementing cover-cropping rotations, increasing permanent and semi-permanent pasture land, adoption of silvicultural and silvopastural systems, expanding riparian zones, afforestation and by 'feeding' soil with carbon-rich amendments [64,81,82].

It is estimated that, if properly managed, carbon farming in Australia could have the potential to remove approximately 497 Mt $CO_2$e $yr^{-1}$; with contributions of approximately 68 Mt $CO_2$e $yr^{-1}$ from arable land, approximately 286 Mt $CO_2$e $yr^{-1}$ from high-volume grazing rangeland and approximately 143 Mt $CO_2$e $yr^{-1}$ from forestry [64,83]. Australia's annual GHG emission has been reported to be 528 Mt $CO_2$e $yr^{-1}$ [84] with agricultural sources contributing 13% of the total GHG emission [73]. Thus, carbon farming in Australia has the potential to completely absolve agricultural GHG emission and, in reality, offset virtually all of Australia's present-day GHG emissions.

The first iteration of the CFI (through the CPM) was a voluntary baseline and credit offset scheme [77]. Where offsets were determined relative to a predefined baseline/reference value, and verified credits were sold or auctioned to ETS-regulated industry, or internationally where recognized as Kyoto Protocol CDM compatible offsets [64,85–87]. Payments were initially made at a carbon floor price of $23 AUD $t^{-1}$ $CO_2$e. To provide an economic disincentive to industry, and encourage divestment from high-emission activities, this price was projected to increase by between 2.5 and 5% per annum [73,80,87]. In its first 2 years of operation (2012–2014), national emissions reduced [84,88], and total emission from the energy generation sector (accounting for approx. 37% of national GHG emission), dropped from 199.1 Mt $CO_2$e $yr^{-1}$ (2012) to 180.8 Mt $CO_2$e $yr^{-1}$ (2014) [74]. However, in late 2014, the CPM (that underpinned offset ETS trading of CFI credits) was repealed [73,80,81]. The repeal and subsequent withdrawal of the CPM was politically motivated by a change in government that negatively framed the CPM as a 'carbon tax' to secure votes [79,87]. Following the withdrawal, Australia's GHG emissions rebounded to exceed 2014 emissions levels (and have done so subsequently each year) [84,88]. Energy sector-specific emissions increased to 187 Mt $CO_2$e $yr^{-1}$ the year following the repeal (2015), and further to 189 Mt $CO_2$e $yr^{-1}$ in 2016 (increasing towards similar levels of emissions from prior to CPM adoption) [74].

In November 2014, the Emissions Reduction Fund (ERF) was established as a successor scheme and granted a budget of $2.55 billion AUD for the following 4 years (2015–2019), and CFI methods were continued [73,80,81,87,89,90]. The ERF operated on the basis of reverse auctioning [73], wherein, CFI projects bid their mitigation/emission-avoidance (i.e. expected quantity of $CO_2$e) and the total operational cost. The most cost-effective schemes are subsequently purchased at auction (in majority by the government, but some by private entities) [73,81]. Although transition to the ERF has led to substantial decreases in the price of carbon (from approx. $23 AUD to approx. $12 AUD $t^{-1}$ $CO_2$e [80]), contracts granted to farmers have been found more economically stable and favourable, providing steady incomes over time [73]. As of October 2020, a total of 866 projects had been registered through the ERF; and more than 85 million credits issued [91].

and biochar [13,31,94,95]. A shift away from aggressive soil tillage regimes (that promote disaggregation of soil, and soil carbon oxidation/mineralization [12,13,85]), to minimum or no-tillage alternatives have reported capacity to rebuild farmland carbon stocks by 0.09–0.12 Gt C in Western Europe yearly [96]. While the use of high carbon soil amendments may improve soil health and deliver long-term

sequestration [31,66,94]. Adoption of such methods to optimize soil management practices could realize annual soil carbon uplifts of 0.6–1.2 Gt C [13,97].

Soil-centric programmes, such as '4p1000' and FAO 'RECSOIL (Re-carbonisation of global soils)' initiatives, have highlighted the opportunity for soils to be at the forefront of global climate change abatement practice and policy [42,66,82,98,99]. By increasing soil carbon stocks in line with methods proposed by '4p1000' (i.e. yearly increases in the carbon content of agricultural soils by 0.4% in the top 40 cm), there is capacity to sequester up to 3.4 Gt C yr$^{-1}$. Such a level of sequestration would provide effective carbon offset for approximately one-third predicted yearly emission from the fossil fuel and cement sectors in 2030 (estimated 10.9 Gt C) [42,66].

# 5. Putting a price on carbon

By assigning a tangible value to a unit of carbon (or more broadly, a unit of $CO_2$e), a mechanism is established that enables charges to be applied to GHG emitters. At present, there are several different carbon valuation metrics, each seeking to place a direct financial, or wider commodified value upon carbon (table 1) [100,105–107]. Under a regime where carbon emission has a 'cost' that can be recovered from a polluter, an economic lever exists to discourage polluting activity and/or encourage operational efficiency and divest from sources of high emission [100,108]. Such a financial instrument provides an economic *disincentive* to continue with current practices, especially in cases where mitigation measures are more financially favourable than business as usual [105,109]. Such a philosophy has its roots in the 'polluter pays principle' that emerged in the 1980s [108,110].

Globally, different regions/nations have taken contrasting approaches to carbon pricing policies, carbon offsetting and carbon trading (electronic supplementary material, table S1) [105]. Current carbon valuation metrics focus heavily upon the aforementioned economic *disincentives*: levying carbon taxes and adoption of emissions trading schemes (ETS)—with yearly reductions in allocated credits (cap-and-trade) [111,112]. These disincentive instruments, although successful at reducing emissions (through fiscal squeeze on emitters), do not ease the burden of carbon already emitted [105]. In many instances, carbon taxes, and carbon trading platforms, have been effective in leveraging business engagement and reducing emissions, while promoting development in low-carbon alternative technologies [100]. Many of these net gains have been associated with ETS (electronic supplementary material, table S1), primarily targeting industry and energy generation sectors that emit large quantities of GHGs [113].

ETS (electronic supplementary material, table S1; table 1 and figure 1) allow for the emission of GHGs to predefined levels, through the allocation or auction of permits that must be 'paid' to the governing body when used (*upon emission of the specified amount of GHG*, generally; 1 permit = 1 t $CO_2$e [109]). The EU currently operates the world's largest ETS [114,115]. Established in 2005, the EU ETS (based on a cap-and-trade mechanism) functions in all EU countries, Iceland, Norway and Lichtenstein. The EU ETS limits emissions from over 11 000 factories, power stations and commercial flights operating between EU member states; and collectively covers around 45% of all EU GHG emissions [116]. The EU ETS has been instrumental in delivering a total reduction of 21% in emissions between 1990 and 2013 [117]. Within the EU ETS, a limited number of emissions permits are directly allocated (based upon the individuals' share of sector emissions, assumed emission from business as usual and calculated sector emission benchmarks [118–120]). Allocations are reduced yearly by 2.2% (post-2021), further encouraging transition and investment into energy efficiency via reduced emission limits and permit scarcity. By extension, permit reductions also lead to increased permit trade in the marketplace and increased permit value, further driving efficiency due to raised operating cost [115,116]. Thus, yearly increases in carbon prices lever increased investment in efficiency and environmentally friendly practice [121]. Remaining permit requirements (where allocations are exceeded) are met through auction and trade at market prices [122–124], or through purchase of equivalent and verified carbon offsets [118]. Emissions permits (and by proxy carbon) have typically been traded between €3 t$^{-1}$ $CO_2$e and €25 t$^{-1}$ $CO_2$e [121].

At present, valid carbon offsets include investment in sustainable or high-efficiency energy generation programmes, and credible certified emissions reductions, primarily sourced internationally (facilitated by the clean development mechanism (CDM) criteria of the Kyoto Protocol) [125–127]. Offsets were limited to 1600 Mt $CO_2$e between 2008 and 2020, due to offset costs being substantially lower than ETS trading prices (and often in third-party countries (*not ETS members*)), thus undermining emissions reductions in favour of paying for cheaper offsets with no direct benefit granted to member states [127,128].

**Table 1.** Disincentive and incentive carbon trading/payment mechanisms.

| | disincentive | | incentive |
|---|---|---|---|
| | emissions trading schemes[a,b] | carbon pricing (taxation)[a,c,d] | carbon offsetting[e] |
| summary | predefined sectors/industries that emit over a certain threshold (of $CO_2e$) must acquire permits in order to operate (1 permit equates to 1 t $CO_2e$). Once obtained, the installation may then operate and emit $CO_2e$ up to the defined permit limit. Additional permits or carbon offsets must be purchased in instances where emissions exceed permit allowance (or fines will be applied). Surplus permits (from under emission) may be sold or auctioned on the ETS market. direct supply of emissions permits are decreased each year to promote scarcity of permits and raise prices, encouraging sustainable development and efficiency | ascribes a price for carbon that may directly tax (all relevant) sources of carbon emission. Payments may be based on the total potential economic, environmental and social cost of emissions or coupled to the carbon market price high price of carbon tax disincentivizes and reduces emissions through increased operating costs | voluntary market-based solution that encourages net emitters of $CO_2$ to buy 'offsets' which may include emission reduction technologies or payment for activities that sequester carbon, thus lowering their emissions by proxy voluntary carbon offsetting can be coupled to ETS schemes or can be paid as stand-alone offsets by individuals or companies |
| valuation mechanism | market price with minimum/maximum boundaries | fixed or market-coupled price | market-based/cost of implementation |
| carbon prices | variable | fixed/variable | variable |
| scale | large companies and industries with emission that exceed the emission threshold | individual—large-scale business and industry | individual—large-scale business and industry (ETS partners) |
| direct reductions in emissions (t $CO_2e$) | yes | yes | no |
| direct payment of sequestration activities | no | no | yes |

[a]Tvinnereim & Mehling [100].
[b]World Bank [101].
[c]Marron et al. [102].
[d]Boyce [103].
[e]Taiyab [104].

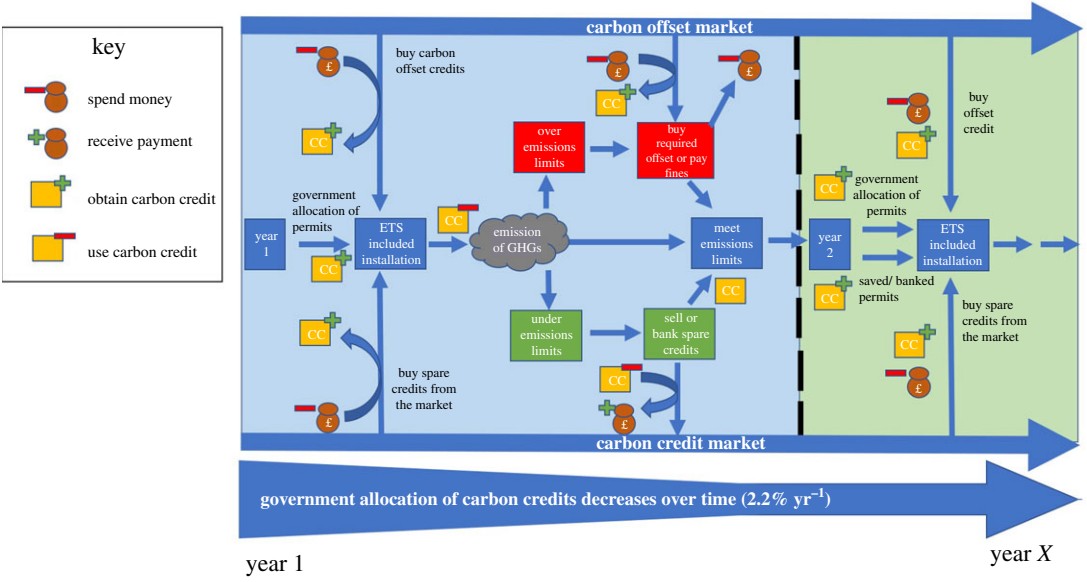

**Figure 1.** Mechanism for increased carbon offset and permit trading within ETS through increased availability and acceptance of verified carbon offsets.

Installations subjected to ETS quotas that exceed yearly emissions caps, and without sufficient additional permits (from verified offsets or purchased in the marketplace), are fined by the ETS governing body. The cost of the fine (approx. €100 t$^{-1}$ CO$_2$e) exceeds market price; thus, an emitter is dissuaded from overrunning their quota [115,116,129]. It is highlighted that a minimum of 50% of the revenue generated through permit auctions and emissions fines is subsequently invested in low-carbon technologies, green-energy projects, environmental protections and sustainable innovation within ETS member states [116,130].

In the UK, carbon trading has historically been operated under the EU ETS umbrella (following guidelines and rules) and affects primarily the energy generation sector [115,131]; however, this has since diverged into a separate UK ETS from January 2021. In addition to ETS market-coupled pricing, a predefined minimum value for which carbon can be traded (a carbon support price) was set in 2013 [132]; an approach that has since been adopted by the EU ETS [133]. The implementation of support prices mitigates issues of permit oversupply or fluctuations in the prices that would destabilize the market [134]. Since 2016, the carbon support price has been frozen at £18 t$^{-1}$ C; however, it is predicted to rise to £30 t$^{-1}$ C after 2023 and further to £70 t$^{-1}$ C by 2030 [115,135]. In the UK emission reductions (catalysed by ETS and linked to divestment from coal power/transition to renewable energy sources) of 77 Mt yr$^{-1}$ from 1990 to 2018 have been reported [67,132].

*Carbon pricing* (table 1) is, an alternative method of disincentivizing emissions, achieved through the use of specialized taxes that charge for carbon emission [136]. Carbon pricing gives a greater flexibility to which goods and services can be taxed directly; thus, offering a non-market-coupled price for carbon [109]. Carbon pricing methods are often not a one size fits all value, and prices are instead adjusted for each good or service, based on the total potential environmental and social costs of the emissions [105,109,137].

Sweden operates direct taxation of carbon emission (alongside EU ETS membership) [100,138]. Sweden's carbon tax is among the highest in the world, with a value of approximately $140 t$^{-1}$ CO$_2$e [100]; following yearly increases since adoption in 1995, when 1 t CO$_2$e was valued at €23 [139]. The scheme operates on the *polluter pays* principle; where the carbon tax is placed upon any activities that emit CO$_2$ (e.g. use of fossil fuels) [108,140], and includes government, industry and private individuals [138]. This carbon tax has facilitated substantial decreases in total emissions, especially within the transportation sector where reductions of 11% were measured (equivalent to 2.5 Mt CO$_2$e [141]) and has culminated in total emissions observed in 2010 to be equivalent to those of 1960, despite continued national growth [138].

Disincentive carbon payment methods often show positive results, such as within the UK, Sweden and wider EU ETS schemes, where large reductions in GHGs have been achieved [67,132]. However,

these schemes are not without their issues. Disincentive payment options are often unpopular, with widespread criticism and political opposition prevalent [126] (box 1). These options are often seen as another form of government-levied tax, where the proceeds go to funding projects unrelated to climate change abatement [125], or through financial squeezing, seen to limit competitiveness and development in the global marketplace [126,142]. Although trade of carbon and direct taxation of emissions facilitate emission reductions, there are often no explicit links to carbon sequestration from the atmosphere. Rather, carbon revenues support a diversity of activities that directly and/or indirectly aspire to deliver lower GHG emissions in the future. Thus, significant steps to reduce or resolve the effects of climate change will only transpire when emissions are reduced, *and* the anthropogenic atmospheric carbon load is re-sequestered concurrently.

Decarbonization of the agricultural sector brings significant opportunity to reduce GHG emissions and re-carbonize soil (§4). Validated offsetting schemes that re-sequester carbon in soils may provide this required level of additionality (sequestration or offset activity that would not otherwise occur) to contemporary carbon markets, over and above low emission investment. Such an approach holds enormous potential to not only rejuvenate soil carbon stocks, but to realize collateral benefits for soil ecosystems and the manifold ecosystem services they support (§2). To achieve this aspiration three key elements are needed: (i) a soil carbon sequestration price (§6), (ii) a soil carbon trading platform (§7), and (iii) assurances on long-term soil carbon storage (§8).

# 6. Establishing a soil carbon price

Given their clear connection to climate change adaptation/mitigation, soil carbon stocks and carbon sequestration have tangible value (§4) [1,37,82]. However, the re-carbonization of soil will underpin manifold benefits for soil health and the delivery of soil ecosystem services (SES) (§2); it is arguably, the holistic value of these outcomes that should be used to establish the soil carbon price. Herein lies the challenge: to accurately value the provision of SES in a way that connects the regulating influence of soil carbon with the direct and wider value it represents to SES provision (including, but not limited to GHG emission mitigation).

The importance of ecosystem service valuation for sustainable growth and development has been recognized since the late 1960s, where the need to include natural resource stocks within decision-making was first identified [3,143,144]. Environmental economic approaches have since been applied to define the *relative* value of ecosystem services [144–146], and more recently, the monetization of ecosystem services has emerged as a useful tool to support payments linked to ecosystem conservation [147,148]. A well-established global exemplar of payments for ecosystem services is the REDD+ scheme that, under an *incentive*-based mechanism, facilitates payments for afforestation and forestry management [149–152].

Encouragingly, dialogue between economists and soil scientists, aligning the economic value of ecosystem services to land use decision-making processes, has gone some way to promoting the ideas of SES valuation and payment [1,3,16,153]. However, with the notable exception of carbon farming (box 1) in Australia, the evaluation and inclusion of payment for soil carbon (or SES) in the mainstream of national/international policy has not yet transpired [16,154]. This circumstance is juxtaposed with the economic costs associated with soil degradation (§3). It has been estimated that the total financial cost associated with soil degradation in the EU exceeds €38 billion yr$^{-1}$, with associated crop loss costing more than €1.25 billion yr$^{-1}$ [155], while in the USA, soil degradation and reductions in soil carbon are estimated to cost at least US \$44 billion yr$^{-1}$ [156]. Globally, the effects of soil degradation compound to a total estimated cost of approximately US \$300 billion each year [157].

The challenge in setting a soil carbon price is not as straightforward as simply linking the soil carbon price to the monetary values associated with the costs of soil degradation. Rather, there is a real need for an expansive and more holistic valuation and assessment that embraces wider SES provision and a broad range of worldviews. Thus, while we recognize that soil carbon markets are a powerful tool to demonstrate the value of soil to decision-makers, we highlight that non-monetary value of soil should, in an ideal world, be promoted concurrently (a view reflected in the Intergovernmental Science-Policy Platform on Biodiversity and Ecosystem Services (IPBES), through its concept of nature's contributions to people [158]). Through a more holistic framing, wider public support for, and adoption of, a soil carbon economy will be facilitated (i.e. from a shared perspective, that everyone will benefit, engagement will be stronger [159]).

Notwithstanding foregone ideals, the overriding proviso remains, soil carbon sequestration prices must square with the carbon market price currently used to facilitate existing carbon trading schemes. Reconciling prices to these already-established carbon markets (table 1) will be essential to mitigate issues pertaining to *offset undermining* of emissions reductions, arising from lower offset costs [127,128], or low uptake on a soil carbon market due to an over-priced soil carbon unit. Furthermore, soil carbon prices must be substantial enough to encourage uptake by farmers and landowners. Stakeholders are only likely to adopt soil carbon sequestration practices where there are valid economic incentives to do so, especially where significant investment in time and resources are involved [64,160]

# 7. A soil carbon trading platform

Assuming a soil carbon price can be ascribed, a platform upon which to trade soil carbon units/credits will be needed to bring a soil carbon economy to fruition. A *voluntary* market (table 1), trading in carbon offsets, could be the way forward to incentivizing payment for those that sequester carbon [161,162]. In 2010, 131 Mt $CO_2$e were traded through voluntary carbon markets; at a total value of US $424 million [163] (estimated increase to 141Mt $CO_2$e traded in 2019). A significant proportion (29%) of the total revenue was associated with the REDD+ carbon market [163]. The REDD+ initiative [149–151] provides financial reward for developing countries that reduce GHG emissions through actions that redress deforestation and forest degradation (facilitated through CDM [127]). Thus, REDD+ incentivizes forest carbon stock improvements, while realizing collateral benefits for sustainable management of forest conservation [149,150]. Voluntary offset markets offer ancillary benefits to farmers and landowners through increased opportunity to diversify production, reduce costs (if following carbon farming methods) and provide new revenue streams [164]. Through monetization of soil carbon sequestration/storage as a soil *good*, these markets can provide win–win opportunities for farmers/landowners, sustainable development and the wider global community [161,163].

Voluntary offset markets hold great potential for increasing the provision of soil (and wider) ecosystem services (§2) [161,165]. It is highlighted that many voluntary offset buyers are also willing to pay higher *premium-offset* rates where wider ecosystem service, biodiversity and societal net gains are delivered in parallel [161,163]. The voluntary carbon market might also facilitate the processing of payments linked to personal or private offsets (i.e. such as is seen from citizen payments to offset GHG emission associated with air travel [166,167]). However, if this course is to be followed, standardization and verification measures must be established to ensure the validity of any subsequent sequestration and offsets that are created (§8). At present, there are several verifying bodies that set and monitor standards and methods within the voluntary offset market (Verra, SCS Global and GoldStandard). These organizations act as a point of registration and verification for projects and offsets credits sold on the voluntary or compliance (ETS) markets, ensuring their validity and additionality. The vast majority of these verified offsets are projects based in the global south and developing nations, focusing on REDD+ in nations such as Brazil, or energy efficiency projects in Kenya keeping costs low. However, for voluntary markets to truly gain traction and effect large-scale change, efforts must be made to better incorporate projects in more developed regions.

Although steps have been taken to increase the scope of offset projects, there is yet to be a mainstream provider or standardizing body focusing on soil-based offsets. An issue we believe pertains to the complexity of soil carbon measurement/monitoring, and the assurity of long-term carbon storage (§8).

As an alternative method, soil carbon payment mechanisms could be integrated into the framework of an existing system: agri-environmental schemes encourage environmentally friendly and sustainable agriculture/land management practice. These incentivized voluntary schemes, providing financial incentives to farmers/land managers to adopt best practice, have been active in the EU since the 1990s [168–170]. Aligning a soil carbon economy with such schemes makes sense, as these schemes fundamentally seek to evoke environmental net gains, many of which link to soil carbon sequestration. In addition, these schemes have pre-existing framework and infrastructure to facilitate relevant payments.

The UK is currently developing new national agriculture regulations (Agriculture Act, 2020) and a new agricultural payment system: Environmental Land Management Scheme (ELMS). Central to developing the policy and payment scheme is the pledge to use public money to pay for public goods [171,172]. Thus, payments are anticipated to follow on farm interventions that deliver public goods, for example, biodiversity net gains. There is also possibility that ELMS could support payments for soil re-carbonization (box 2).

**Box 2.** A soil-centric approach—case study UK.

As the UK withdraws from the EU, it will need a UK-specific successor to current EU agri-environmental schemes. The UK Government, through the Department for Environment, Food and Rural Affairs (DEFRA), is in the process of creating a UK-centric Environment Land Management Scheme (ELMS). The development of ELMS coincides with the adoption of the new Agriculture Act (2020), and Environment Bill [173]. ELMS will provide a mechanism through which to meet Net Zero targets set by DEFRA and the National Farmers' Union (NFU), by, respectively, 2050 and 2040 [70,160,172,174]. ELMS will be centred on the philosophy of using 'public money to pay for public goods' [171,173,175]. It is ELMS' mission to deliver manifold environmental benefits by providing farmers, foresters and other land managers with opportunities, incentives and financial reward for enhancing or maintaining the environment and essential ecosystem services while protecting UK natural capital [171,173,175].

ELMS will provide a three-tiered management scheme. Payments will be made to farmers and landowners for the ecosystem services provided, rather than payments based on total farm area, livestock herd size and general environmentally sensitive practices as seen previously [173]. The public goods that will probably be paid for under ELMS include the provision of clean and plentiful water, clean air, protection from and mitigation of environmental hazards (e.g. flooding), mitigation of and adaptation to climate change, thriving plants and wildlife, habitat protection and expansion, beauty, heritage and engagement [171,172].

The payment mechanisms under which ELMS will operate are currently under discussion but are likely to include instruments such as: government-based price setting (fixed prices), market-coupled price setting (linking to the commodified value of carbon, allowing private sector investments and offset payments similar to ETS), direct payment mechanisms or payment by results, where a portion or all of the payment is delayed until adequate benefit has been attained [171].

ELMS is an evolving payment system, and while no formal pledge has been made to incorporate payment to farmers and landowners to sequester carbon in soils such an outcome could transpire [70,171–173]. Payments to support soil carbon sequestration could be implemented as a component of ELMS or, perhaps more likely, could be developed in parallel to ELMS under a voluntary offset scheme. ELMS is currently in the planning and trial stage (2021–2024) prior to full national adoption post 2024. It is likely, in the interim, that pressure will be placed on the government to instate 'carbon farming' policies into ELMS before the full roll out of the scheme.

Linking together the Agricultural Act (2020) and the Environmental Bill to ELMS, alongside the NFU Net Zero 2040 aspiration, will ensure agricultural profitability and sustainability in the UK, enhancing the environment and working towards climate change goals in an effective manner [173]. Payment for re-carbonization of soils, through ELMS, would help catalyse the transition of the UK agricultural sector to Net Zero, and provide opportunities for enhanced delivery of ecosystem services [14,17,18]. Such a soil-centric approach would provide a fast-track to optimizing the delivery of public goods and services, while cementing and improving profitability of the agricultural sector in the UK.

A recent addition to the private sector carbon trading marketplace is the US-based company IndigoAg. Following release of their Terraton Initiative in 2019, the company has vowed to initiate its own carbon offset platform. The initiative aspires to the goal of sequestering 1 Tt (1 Tt = $10^{12}$ tonnes) of C globally (IndigoCarbon) via incentivized carbon farming (box 1). To facilitate this aspiration IndigoAg propose to offer a minimum carbon price of US $15–20 t$^{-1}$ CO$_2$e sequestered, such a value corresponds to that proposed as the 'feasible minimum carbon price' for significant and effective carbon sequestration in farmland soils [160]. At the time of writing, the UK Government had recently announced its 2021 Budget Statement [135], contained within which was an aspiration to grow a *green taxonomy*, with the UK at the centre of an expanded global voluntary carbon market.

While sequestration projects (such as REDD+) and Terraton are recognized as effective carbon offset methods (facilitated through the CDM compliance markets and the voluntary market, respectively) [151,162,176] (§5), there has been limited uptake and trade of these offset permits within more formal carbon trading schemes (i.e. EU ETS (§5)). There are three primary reasons for this reticence; firstly, the complexity of the environmental economics involved in sequestration accountancy makes auditing

lengthy, bureaucratic and difficult; secondly, lack of standardization in carbon measurement; and thirdly, uncertainty regarding the permanence of carbon sequestration [162,176–178].

## 8. Assuring long-term soil carbon storage

It would be inappropriate to make payments for short-lived uplifts associated with labile/degradable carbon, on the auspices of GHG mitigation/offset. Payments for soil carbon sequestration should be linked to interventions that deliver *long-term* carbon storage. However, to assess changes through direct measurement of SOC every time an intervention is made would be too costly and time-consuming to be pragmatic [42]. Thus, numerical modelling is an essential tool to deliver confidence in soil-based carbon sequestration potential in a cost-effective way. SOC simulation models are varied but are generally based on empirical relationships or underlying processes, established using long-term field experiments as a primary data source [179,180]. SOC models predict SOC dynamics regionally and globally in response to climatic changes, land use and land management [181–183]. SOC models have been successfully used to predict the impact of agricultural activities on SOC and $CO_2$ emissions, allowing farmers and regulators to predict SOC storage and stability in implementing and developing suitable land management options (e.g. soil amendment application) [179,184].

Here, several models have been developed and are well established for predicting SOC turnover in agricultural soils, for example, the *RothC Model* [185,186], *CENTURY* [187] and *ICBM* [188]. Each model considers the SOC held in different pools with varying decomposition rates; the temporal dynamics of carbon leaving/entering these pools then propagates through, for increasing timeframes. Significantly, the *RothC,* and other models, have been calibrated with measured data drawn from long-term experiments [31,181,182,189] and have been modified to predict the fate of exogenous organic inputs (e.g. compost, agri-industrial waste and digestate) [190–192]. However, further validation is still required to ensure predicted carbon sequestration potential is corroborated/verified for soils and climates directly relevant to locations where soil re-carbonization is delivered.

Verification and validation processes need to be transparent if incentives for land-based carbon sequestration are to be made credible [161]. Placing focus on improving verification and validation processes will save time and expense, thus lowering the barrier for entry, increasing uptake and developing further environmental economic potential for soil re-carbonization. In our view, only with tailored assessment of carbon stability prognoses (that lock to specific bioclimatic regimes, land use and specific interventions) can payments be appropriately reconciled with soil re-carbonization.

## 9. Outlook: soil carbon payments and multiple net gains

Existing carbon trading mechanisms have highlighted the enormous potential for economic levers to deliver sizable reductions in GHG emissions (§4). Furthermore, it is expected that in the short to medium term commodification of carbon will continue to gain traction, through increasing adoption of carbon taxes, expansion of ETS (table 1; electronic supplementary material, table S1), validated carbon pricing, payment for carbon offsets (§6) and the alignment of private sector markets (§7) [105,107]. While increased delivery of disincentive carbon payments will facilitate greater efficiency and encourage divestment from high-emission activities [193–195], it will not redress the fundamental problem of elevated GHG loads already in the atmosphere. To grasp a Net Zero future, historic anthropogenic carbon emissions must also be removed. Thus, it is our view that incentivized carbon payment metrics (linked to sequestration of carbon from the atmosphere) are needed, as a mainstream compliment to these more dominant disincentive (emissions reduction) metrics.

Given the large historic transfer of carbon from soil to the atmosphere (§3), re-carbonization of soil makes intuitive sense. Re-carbonizing soils can provide efficient and cost-effective carbon sequestration potential (§4), without required development of new technology or techniques, and can be applied at scale with relative ease [42,66,82]. Furthermore, through rejuvenation of soil carbon stocks, benefits for soil health and optimization of ecosystem services can be achieved (§3): a win–win outcome.

Assigning a price for *soil* carbon (§6), within the prevailing carbon economy, offers enormous potential to not only combat climate change via economic leverage [196], but also achieve wider societal economic net gains through uplifted delivery of ecosystem services [16,48,161,163,197]. Carbon sequestration payments may also provide opportunity to assist in the sustainable development of underperforming and industrializing regions, contributing to sustainable development goals through payment for beneficial land management practice [48,197,198]. However, setting a soil carbon price is

not trivial. Proportionate payment for carbon sequestration must be established to reconcile economic difficulties faced by farmers/land managers in achieving sequestration aspirations.

With the projected increases in both the price of carbon and adoption of carbon pricing initiatives, financial incentives to sequester carbon will intensify [198]. Herein lies opportunity to formalize a soil carbon economy while the carbon market is in its formative stage (§7). A market-coupled approach would see steady increase in the carbon price through successive yearly increases in the value of emissions permits, simultaneously providing a source of offsets that may be used in conjunction with emissions reductions (figure 1). Setting minimum carbon payment levels and price floors within these adopted schemes (between \$15 and \$20 $t^{-1}$ $CO_2$e) would provide adequate economic incentive to sequester carbon in soil and rectify many of the economic difficulties faced by farmers [160]. Such an approach of integrating sequestration payments into current carbon markets would, however, need to address issues of *lowest cost purchasing*. Specifically, offsets must not be valued at a substantially lower price than emissions credits, effectively encouraging emitters to buy cheap offsets rather than curb emissions. Assuming a price balance can be achieved, an approach that requires incentive–disincentive linkage could catalyse a ground-shift that would reduce *and* mitigate emissions, actively addressing climate change issues.

Interventions to which payments are linked will also require clear articulation (§6). These elements, while procedural, are arguably the greatest challenge to address. With appropriate and proportional political momentum, it will be possible to encourage swift adoption and wide-scale participation in a soil carbon economy by multiple stakeholders (including farmers and landowners, scientists, economists and policy-makers). Specifically, for global re-carbonization of soil to be realized, communication defects and gaps, arising from the convergence of disparate fields (i.e. ecology, economics, agriculture, soil science and governmental policy), must be reconciled [3,16,199]. Monetary valuation of ecosytem services offers potential here. However, we continue to recognize the importance of non-monetary valuation of soil, and that this should be accommodated in soil carbon price setting.

Furthermore, concensus needs to be reached on how carbon stocks are defined, and stock changes verified through (i) SOC measurement and (ii) SOC durability. With regard to these aspects, international agreement is needed to define the depth to which carbon stocks should be assessed, and in what form the carbon (labile/recalcitrant) is considered 'eligible' for remuneration. In considering the durability of a carbon stock, agreement will be required regarding the timeframe of the sequestration prognosis and standardization of SOC fate-model variables (§8). Much of the uncertainty surrounding soil carbon markets can be mitigated by drawing upon contemporary literature, corroborating the environmental and economic value of soil carbon sequestration. It is our view, however, that this be tempered with a pro-active approach, where evidence may be gathered and synthesized through ongoing action/intervention—actively engaging with soil carbon payments.

In conclusion, the re-carbonization of soils has the capacity to deliver the required intervention to offset current emissions and remove historic emissions. Furthermore, re-carbonization of soil will deliver climate change mitigation with co-benefits for soil quality, health, the delivery of SES and wider societal benefits. Capturing a *soil-based* carbon economy is a grand challenge, but with urgent and assertive political action, one that is attainable in the decade ahead.

Data accessibility. As this is an Evidence synthesis paper, no new data have been presented herein. The authors have, however, appended electronic supplementary material that may be accessed to emphasize relevant portions of the manuscript. References for such supplementary materials are included in the manuscript reference list and are cited below: [100–104].

Authors' contributions. S.G.K. and B.J.R. led the drafting of the concept paper. Thereafter, A.F.R., L.M., A.E.L. and A.R.H. provided text to develop specific sections aligned to their expertise. The entire team was involved in review and editing to deliver the final manuscript.

Competing interests. B.J.R. and A.E.L. are editors to the special collection on sustainable land use, and as such, a different handling editor is required.

Funding. This research was supported by the Research England's Connecting Capability Fund (CCF; through the EIRA projects VALCRUM and PANEZA), The Royal Society (Advanced Fellowship Award NAF\R2\180676), Carlos Chagas Foundation for Research Support of the State of Rio de Janeiro Faperj (E-26/202.680/2018), Coordenação de Aperfeiçoamento de Pessoal de Nível Superior Finance (code 00) and National Council for Scientific and Technological Development—CNPq (308536/2018-5; for the project 'Sustaining the land from the ground up: developing soil carbon and soil ecosystem services valuation frameworks for tropical soils').

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
