## [Peer Review File · Royal Society Open Science]

Review History

RSOS-202305.R0 (Original submission)

Review form: Reviewer 1

Is the manuscript scientifically sound in its present form?

Yes

Are the interpretations and conclusions justified by the results?

Yes

Is the language acceptable?

Yes

Do you have any ethical concerns with this paper?

No

Have you any concerns about statistical analyses in this paper?

No

Recommendation?

Accept with minor revision (please list in comments)

Comments to the Author(s)

This is an excellent review on the carbon issue in general, and specifically on carbon in the soil. It brings, on one hand, a state of the art view on the carbon cycle in the soil and, on the other hand, as social and low-carbon economy policies. The language is clear and the text is pleasant to read. Some minor recommendations and doubts:

In line 66: "The soil was treated as a lifeless constant". Treated by whom?

Line 121: "Soils with high levels of aggregation and structure are better equipped to accommodate rainfall". "More suitable" sounds better

In line 153: Where soil loss / degradation rates exceed soil replacement / recovery rates, the balance of sustainability is tilted. As it is an interdisciplinary work, it is important to briefly inform the reader about the processes and rates of soil replacement / recovery. Are you referring to geological substitution, biological or both?

In line 369: I suggest that you highlight that carbon fixation in the soil brings ecological benefits also on a very local scale (as in soil fertility and erosion control). In other forms of C fixation, benefits generally occur on a more regional or even in a global scale.

Review form: Reviewer 2

Is the manuscript scientifically sound in its present form?

No

Are the interpretations and conclusions justified by the results?

No

Is the language acceptable?

Yes

Do you have any ethical concerns with this paper?

No

Have you any concerns about statistical analyses in this paper?

No

Recommendation?

Major revision is needed (please make suggestions in comments)

Comments to the Author(s)

The manuscript presents consistent information on the importance of the carbon stock and compensation mechanisms, pointing out case studies and paths, having scientific merit and potential to support decision and policies. However, it is too long for this kind of publication, taking into account its potential to subsidize decision and policy makers focused on the theme. There are gaps in terms of defining the focus of manuscript and does not meet the format required for an Evidence Synthesis.

Decision letter (RSOS-202305.R0)

Dear Mr Keenor

The Editors assigned to your paper RSOS-202305 "Capturing a Soil Carbon Economy" have now received comments from reviewers and would like you to revise the paper in accordance with the reviewer comments and any comments from the Editors. Please note this decision does not guarantee eventual acceptance.

Please submit your revised manuscript and required files (see below) no later than 21 days from today's (ie 16-Feb-2021) date. Note: the ScholarOne system will 'lock' if submission of the revision is attempted 21 or more days after the deadline. If you do not think you will be able to meet this deadline please contact the editorial office immediately.

on behalf of Professor Pete Smith (Subject Editor)
openscience@royalsociety.org

Associate Editor Comments to Author (Professor Pete Smith):

Both reviewers recommend revisions, with reviewer 2 having more critical comments. Please address these comments.

Reviewer comments to Author:

Reviewer: 1

Comments to the Author(s)

This is an excellent review on the carbon issue in general, and specifically on carbon in the soil. It brings, on one hand, a state of the art view on the carbon cycle in the soil and, on the other hand, as social and low-carbon economy policies. The language is clear and the text is pleasant to read.

Some minor recommendations and doubts:

In line 66: "The soil was treated as a lifeless constant". Treated by whom?

Line 121: "Soils with high levels of aggregation and structure are better equipped to accommodate rainfall". "More suitable" sounds better

In line 153: Where soil loss / degradation rates exceed soil replacement / recovery rates, the balance of sustainability is tilted. As it is an interdisciplinary work, it is important to briefly inform the reader about the processes and rates of soil replacement / recovery. Are you referring to geological substitution, biological or both?

In line 369: I suggest that you highlight that carbon fixation in the soil brings ecological benefits also on a very local scale (as in soil fertility and erosion control). In other forms of C fixation, benefits generally occur on a more regional or even in a global scale.

Reviewer: 2

Comments to the Author(s)

The manuscript presents consistent information on the importance of the carbon stock and compensation mechanisms, pointing out case studies and paths, having scientific merit and potential to support decision and policies. However, it is too long for this kind of publication, taking into account its potential to subsidize decision and policy makers focused on the theme.

There are gaps in terms of defining the focus of manuscript and does not meet the format required for an Evidence Synthesis.

The manuscript objective was presented in a diffuse way, not allowing identifying clearly whether its goal is to summarize evidence in a policy-neutral manner, or whether it is making the case for a particular course of action based on expert interpretation of the underlying evidence, item required to publish an Evidence Synthesis.

The literature review (importance of soils, carbon stocks, ecosystem services, re-carbonisation of soil, etc.) was quite extensive, more appropriate for a Paper Review.

The focus here could be on strategies and tools to operation of carbon markets (public and private sectors) that consequently will combat soil degradation and climate change.

There is a need for a spelling check.

I still suggest focusing on a single compensation mechanism because the strategies and tools for Payment for Ecosystem Services schemes are completely different from those aimed at the carbon market.

The review is also diffuse and dealing with various subjects such as numerical modeling to estimate SOM, gaps in its monitoring, etc. I think the focus should be on strategies, tools and other aspects related to the carbon credit market, since there is a proposal of a soil carbon trading platform.

Thus, the authors could bring the case studies (into the boxes) from other countries that already have this market established, pointing out successes and gaps and how they could be supplied in Brazil.

Thus, the manuscript would meet the requirements to an Evidence synthesis:

1) Highlight priority issues and the need for evidence to inform decisions, typically involving evaluation of published information in relation to needs, and sometimes interactions across organizations.

2) Critical matching of need to response within a policy context is crucial not only to secure timeliness, but also to identify priorities in the face of limited resources. In such circumstances, any such decisions require the best available evidence.

On the other hand it was not so clear how this platform would work, I imagine that based on Figure 1 (proposed by the authors), which in my view was excellent and should be the heart of the publication.

===PREPARING YOUR MANUSCRIPT===

===PREPARING YOUR REVISION IN SCHOLARONE===

Please ensure that you include a summary of your paper at Step 2 'Type, Title, & Abstract'. This should be no more than 100 words to explain to a non-scientific audience the key findings of your

research. This will be included in a weekly highlights email circulated by the Royal Society press office to national UK, international, and scientific news outlets to promote your work.

<https://royalsociety.org/journals/authors/author-guidelines/#supplementary-material> to include a suitable title and informative caption. An example of appropriate titling and captioning may be found at https://figshare.com/articles/Table_S2_from_Is_there_a_trade-off_between_peak_performance_and_performance_breadth_across_temperatures_for_aerobic_sc_ope_in_teleost_fishes_/3843624.

Author's Response to Decision Letter for (RSOS-202305.R0)

See Appendix A.

Decision letter (RSOS-202305.R1)

Dear Mr Keenor,

I am pleased to inform you that your manuscript entitled "Capturing a Soil Carbon Economy" is now accepted for publication in Royal Society Open Science.

You can expect to receive a proof of your article in the near future. Please contact the editorial office (openscience@royalsociety.org) and the production office (openscience_proofs@royalsociety.org) to let us know if you are likely to be away from e-mail contact – if you are going to be away, please nominate a co-author (if available) to manage the proofing process, and ensure they are copied into your email to the journal. Due to rapid publication and an extremely tight schedule, if comments are not received, your paper may experience a delay in publication.

on behalf of Professor Pete Smith (Subject Editor)
openscience@royalsociety.org

Editor Comments to Author (Professor Pete Smith):
The reviewer comments have been addressed in a satisfactory way, and the manuscript does not need to be re-reviewed

Appendix A

9 March 2021

Dear Editor,

Manuscript ID RSOS-202305, "*Capturing a Soil Carbon Economy*"

We are grateful to the reviewers for their comments. Changes have been made to the paper in accordance with the comments received and as described in the actions tabulated below. We have tracked changes within the document for ease of review ("tracked version") and provided a "clean version" with changes accepted.

In making our revision we have reduced the manuscript by 1 page (Reviewer 2). In addition, we have made revision to emphasise *support for* a soil carbon market that will encourage the re-carbonisation of soil. We are not neutral (Reviewer 2) and have made our position clearer.

We hope that the changes made are acceptable.

Sincerely,

Brian Reid

Reviewer	Line in text	Comment	Action
1	66	"The soil was treated as a lifeless constant". Treated by whom?	Sentence has been removed.
1	121	"Soils with high levels of aggregation and structure are better equipped to accommodate rainfall". "More suitable" sounds better	Word changed and sentence edited.
1	153	Where soil loss / degradation rates exceed soil replacement / recovery rates, the balance of sustainability is tilted. As it is an interdisciplinary work, it is important to briefly inform the reader about the processes and rates of soil replacement / recovery. Are you referring to geological substitution, biological or both?	Sentence edited for greater clarity of soil replacement origins, in line with reviewer's comments on geological and biogenic soil origin.
1	369	I suggest that you highlight that carbon fixation in the soil brings ecological benefits also on a very local scale (as in soil fertility and erosion control). In other forms of C fixation, benefits generally occur on a more regional or even in a global scale.	Comment to this effect added i.e. carbon benefits across scale.
2	Whole doc	Paper is too long, should be abridged	Edits have been made throughout the paper to abridge and make literature more concise. However, in line with addressing some of the reviewers comments additional content has been added. Overall, the paper has been cut by 1 page in length. At a total of 21 pages (double spaced 12 Pt Times) the paper is, in our view, a standard length. Wordcount of main text + abstract = 6220
2	Whole doc	The manuscript objective was presented in a diffuse way, not allowing identifying clearly whether it goal is to summarize evidence in a policy-neutral manner, or whether it is making the case for a particular course of action based on expert interpretation of the underlying evidence, item required to publish an Evidence Synthesis. The literature review (importance of soils, carbon stocks, ecosystem services, re-carbonisation of soil, etc.) was quite extensive, more appropriate for a Paper Review. The focus here could be on strategies and tools to operation of carbon markets (public and private sectors) that consequently will combat soil degradation and climate change.	In line with reviewers comments the abstract and introduction have received edits to better highlight and clarify better the scope and purpose of the paper. We do indeed seek to support the re-carbonisation of soil. In making revisions we have made this direction more obvious.

2	Whole doc	There are gaps in defining the focus of the MS	See reply above
2	Literature review	Too extensive for this type of article	Literature review has received edits and is now more concise. However, we highlight that we have prepared the manuscript to make the topic accessible across interest groups. While, for example, the text on soil ecosystem services might be prosaic to the soil scientist this is valuable context for the environmental economist reading the manuscript. We have in our editing sought to reduce the length of the paper (see tracked version). We further highlight that sections are cross-referenced throughout. This allowing a reader to target their reading where they seek specific insight.
2	Whole doc	The review is also diffuse and dealing with various subjects such as numerical modelling to estimate SOM, gaps in its monitoring, etc. I think the focus should be on strategies, tools and other aspects related to the carbon credit market, since there is a proposal of a soil carbon trading platform.	Our view is not for the paper to be a blueprint for what a carbon market should be or look like. Rather we highlight the need for this market to facilitate climate goals. We feel that text relating to the actual operational activity of putative soil carbon market would be to speculative and would undermine credibility in the manuscript. Furthermore, adding this text would lengthen the paper – the reviewer is seeking a shortening of the paper. No action taken
2	Whole doc	Need for a spelling check	Manuscript has been revised and reviewed thoroughly for UK spelling.
2	Payment for ecosystem services	I still suggest focusing on a single compensation mechanism because the strategies and tools for Payment for Ecosystem Services schemes are completely different from those aimed at the carbon market.	The existing carbon market landscape is complex and multifaceted. To focus only on the ETS or on the Voluntary Market would be misleading. In pulling the text together we seek to give the non-expert insight to the remuneration options in the carbon market space. A qualifier in this regard has been added to the MS.
2	Modelling	Too diffuse and out of place – focus should be placed on strategies and aspects of the soil carbon market due to proposal of trading and market	In preparing the MS we thought long and hard on the potential bottleneck that might limit the emergence and growth of a soil carbon market. It is our view that for surety in long-term carbon storages is imperative to credibility of soil carbon credits and consumer confidence. Here there are two

			options: measure carbon or model it. In revising the MS we assert why we feel modelling is a preferred option. We feel the modelling section is essential as it highlights one of the key issues within creation of a market and wider use of soil sequestration due to uncertainties. This is also noted and drawn upon through other sections of the paper and backed up with important and valid references. i.e., Smith et al 2020. Ultimately certified/verified soil carbon credits will be the pivot to enable the development of soil carbon trading.
2	Boxes	Thus, the authors could bring the case studies (into the boxes) from other countries that already have this market established, pointing out successes and gaps and how they could be supplied in Brazil.	We unaware of any soil carbon relevant market mechanisms that currently operate in the mainstream. In selecting case studies for the boxes the objective was not to create a large collection of case studies and review, but instead to pick out two contrasting, and topical, carbon payment regimes. One well established, and the other emerging. In accommodating the Reviewer, we have added a paragraph (in the main text) regarding Verra (a global carbon credit registrar); and allude to its voluntary carbon off set operations in Brazil.
2	PRIORITY	Highlight priority issues and the need for evidence to inform decisions, typically involving evaluation of published information in relation to needs, and sometimes interactions across organizations.	In revising the MS we have added conclusion material regarding this comment. Specifically, we draw attention to the fact that national and international shifts to accommodating a soil carbon economy will need to be evidence based. Some of this evidence can be gleaned from the literature available now, while further evidence will need to be gathered and synthesised based on ongoing action/intervention. We also emphasise the distinction between some current off-set approaches that mitigate or abate carbon emissions/loss vs the need to become more proactive in the re-carbonisation of soil.

2	PRIORITY	Critical matching of need to response within a policy context is crucial not only to secure timeliness, but also to identify priorities in the face of limited resources. In such circumstances, any such decisions require the best available evidence. On the other hand it was not so clear how this platform would work, I imagine that based on Figure 1 (proposed by the authors), which in my view was excellent and should be the heart of the publication.	See comment above Re evidence. More focus and effort could be placed on outlining HOW a market may work and become established within a global context, this would however require a fair bit or re-tailoring the paper which in my opinion loses sight of what we are actually discussing and shifts focus more towards a policy recommendation rather than a review or synthesis that highlights the missing aspects of soil carbon and ecosystem service valuation. In discussion of creating figure 1 we highlight that this framework, while designed for carbon per se, would be well aligned with soil carbon markets. However, the ETS divests its income across a range of interventions and in reality, only a few of these actually sequester carbon. In revising our discussion, we highlight this and the counterpoise that a soil carbon market would be better tethered to active soil re-carbonisation.
2	Figure	Figure 1 should play a far more centralised role in the paper.	See above.